# A New Probiotic Formulation Promotes Resolution of Inflammation in a Crohn’s Disease Mouse Model by Inducing Apoptosis in Mucosal Innate Immune Cells

**DOI:** 10.3390/ijms252212066

**Published:** 2024-11-10

**Authors:** Carlo De Salvo, Abdullah Osme, Mahmoud Ghannoum, Fabio Cominelli, Luca Di Martino

**Affiliations:** 1Department of Pathology, School of Medicine, Case Western Reserve University, Cleveland, OH 44106, USA; cxd198@case.edu (C.D.S.); fabio.cominelli@uhhospitals.org (F.C.); 2Department of Anatomic Pathology, University of Alabama at Birmingham, Birmingham, AL 35294, USA; aosme@uabmc.edu; 3Center for Medical Mycology and Integrated Microbiome Core, Department of Dermatology, University Hospitals Cleveland Medical Center, Case Western Reserve University, Cleveland, OH 44106, USA; mag3@case.edu; 4Case Digestive Health Research Institute, School of Medicine, Case Western Reserve University, Cleveland, OH 44106, USA; 5Department of Medicine, School of Medicine, Case Western Reserve University, Cleveland, OH 44106, USA

**Keywords:** probiotics, Crohn’s disease, apoptosis, digital spatial profiling, amylase

## Abstract

The interaction between gut-residing microorganisms plays a critical role in the pathogenesis of Crohn’s disease (CD), where microbiome dysregulation can alter immune responses, leading to unresolved local inflammation. The aim of this study is to analyze the immunomodulatory properties of a recently developed probiotic + amylase blend in the SAMP1/YitFc (SAMP) mouse model of CD-like ileitis. Four groups of SAMP mice were gavaged for 56 days with the following treatments: 1) probiotic strains + amylase (0.25 mg/100 µL PBS); 2) only probiotics; 3) only amylase; PBS-treated controls. Ilea were collected for GeoMx Digital Spatial Profiler (DSP) analysis and histological evaluation. Histology assessment for inflammation indicated a significantly reduced level of ileitis in mice administered the probiotics + amylase blend. DSP analysis showed decreased abundance of neutrophils and increased abundance of dendritic cells, regulatory T cells, and macrophages, with a significant enrichment of five intracellular pathways related to apoptosis, in probiotics + amylase-treated mice. Increased apoptosis occurrence was confirmed by (TdT)- deoxyuridine triphosphate (dUTP)-biotin nick end labeling assay. Our data demonstrate a beneficial role of the probiotic and amylase blend, highlighting an increased apoptosis of innate immunity-associated cell subsets, thus promoting the resolution of inflammation. Hence, we suggest that the developed probiotic enzyme blend may be a therapeutic tool to manage CD and therefore is a candidate formulation to be tested in clinical trials.

## 1. Introduction

The gastrointestinal tract consists of a complex ecosystem in which the constant exposure to multiple microbes induces unique responses in the immune subsets residing in the gut [1]. Recent studies have indicated that the interaction between the mycobiome (fungal community) and the bacteriome (bacterial community), often referred to as the microbiome, plays a critical role in the pathogenesis of inflammatory bowel disease (IBD) [2,3], particularly causing the production of polymicrobial biofilms which in turn allow an uncontrolled microbial growth in the gut, giving pathogens multiple advantages such as antibiotic resistance [4], protection against the host immune system [5], and increased surface adhesion and ability to damage the epithelial gut lining [6]. The resulting dysbiosis caused by the unsuppressed microbial pathogens’ growth is particularly relevant in IBD pathogenesis, as innate and adaptive immune responses can be altered by the microbiome dysregulation, leading to unresolved local inflammation [7].

Besides this detrimental impact that gut-residing microorganisms can have on the host health, there are specific microbial communities that instead play a pivotal role in maintaining gut homeostasis by modulating immune cells responses. Such responses help in controlling the tolerance to commensal microbes, involving both adaptive and innate immunity [8], leading to protective effects [9]. The beneficial effects achieved by this “immune cells–commensal bacteria” interaction induces boosted immunologic responses particularly towards inflammatory [10] and infectious conditions [11].

As such, it is reasonable to suggest that the administration of beneficial microorganisms (probiotics) may help to ameliorate the gastrointestinal manifestations in IBD patients by modulating immune responses and therefore contributing to resolving chronic inflammation. As specified by the World Health Organization, probiotics are defined as live microorganisms which, when administered in adequate amounts, confer health benefits to the host [12]. Probiotics are represented by various bacterial and fungal strains that, when administered at a functional dose, can be used as supplements in the general population [13], positively affecting the gut microbiome composition and interacting with diverse immune cell subsets, thus strengthening the response of the immune system [14] and preventing the progression of multiple diseases, such as cancer [15] and obesity [16]. Therefore, probiotics may represent a cost-efficient alternative solution in regard to managing multiple types of pathologies [17,18].

Although multiple studies have demonstrated probiotics’ beneficial effects, particularly in irritable bowel syndrome [19], the molecular mechanisms driving the interaction between probiotics and the immune system that lead to immunomodulatory effects remain unclear.

We have previously shown that a recently developed probiotic blend containing bacterial and fungal strains (*Lactobacillus* (*L*.) *rhamnosus*, *L*. *acidophilus*, *Bifidobacterium* (*B*.) *breve*, and *Saccharomyces* (*S*.) *boulardii*) coupled with the hydrolytic enzyme amylase is capable of inhibiting the germination of pathogenic biofilm-producing microbial species in vitro [20] and ameliorates CD-like ileitis in a SAMP1/YitFc (SAMP) mouse model. The SAMP mouse strain develops CD-like ileitis spontaneously, without chemical or immunological manipulation [21]. One of the main factors causing attenuation of ileal disease is represented by the probiotic’s ability to increase β-diversity and avoid dysbiosis in the gut [22].

The probiotic and amylase combination has, in fact, been shown to have a positive effect by altering the intestinal microbiome and avoiding dysbiosis. However, the mechanisms underlying its interaction with specific intestinal immune cell populations have not yet been fully elucidated. Thus, in the present study, we investigated how the probiotics and amylase blend affects the distinct immune subsets present in the gut of SAMP mice and what intracellular pathways are particularly induced or inhibited. 

Herein, we show that the probiotics and amylase are both necessary to decrease ileitis in SAMP mice, as the experimental groups administered only amylase or only probiotics showed higher levels of inflammation, similar to the ileitis present in the vehicle-treated control group. Mechanistically, we report that the combination of amylase and probiotics is able to promote significant alterations in distinct immune cell subsets resident in the intestinal mucosa, particularly affecting dendritic cells, innate lymphoid cells (ILC)s, macrophages, regulatory T cells (Treg)s, and neutrophils. Finally, our data suggest that amylase anti-biofilm activity promotes probiotics infiltration into the intestinal mucosa, triggering apoptosis of some innate immune cell populations present in the lamina propria, such as neutrophils, thus facilitating resolution of inflammation in the ileum.

In conclusion, our data support the benefit of testing the developed probiotic-amylase blend as a therapeutic supplement in clinical trials focused on the management of IBD symptoms.

## 2. Results

### 2.1. Amylase and Probiotics Are Both Necessary to Ameliorate Inflammation in SAMP Mice

First, we tested the ability of the amylase alone or probiotic strains alone to decrease the inflammation in the ileum of SAMP mice by oral administration. Histological analysis revealed that the only effective treatment in ameliorating ileitis was the amylase combined with the probiotic strains when compared with the other three groups (Figure 1A), as indicated by the ileal tissue presenting better-preserved villi architecture and attenuated transmural and active inflammation (Figure 1B).

### 2.2. Digital Spatial Profiling of Genetic Expression Performed on Mucosal Ileal Tissue in SAMP Mice

We analyzed the transcriptome profile within distinct Regions of Interest (ROI)s (N = 36/group) in the mucosa of ileal tissues of the four experimental groups employing Digital Spatial Profiler (DSP) analysis using the NanoString platform (Figure 2A). First, we investigated the mucosal layers selected in each ROI using the GeoMx Mouse Whole Transcriptome Atlas assay (which encompasses 22,000 genes), and then we applied spatial transcriptomics to reveal the association between tissue architecture and immune response pathways. Representative histological sections of ilea employed for GeoMx analysis indicating CD45^+^ cells (red), pan-cytokeratin (PanCK)^+^ cells (yellow), and nuclear staining (green) are shown in Figure 2B.

### 2.3. Probiotics Mix Coupled with Amylase Exerts an Enhanced Immunomodulatory Effect in the Mucosal Layer

The biological clustering of ROIs, as visualized by a Principal Component Analysis (PCA) dimensional reduction plot, highlighted a clear separation of the different ROIs belonging to the probiotic + amylase group based on the CD45^+^ marker (Figure 3A). Furthermore, we performed whole-transcriptome analysis of the ROIs to characterize the immune cell populations and then performed deconvolution employing the SpatialDecon R library to evaluate immune cell abundance [23]. The deconvolution revealed a remarkable cluster of the ROIs belonging to the “probiotic + amylase” treatment group (Figure 3B). Among the analyzed immune cell populations, neutrophils were markedly reduced with the probiotics + amylase blend treatment compared to all the other experimental groups. On the other hand, dendritic cells, ILCs, macrophages, and Tregs were consistently increased in the probiotic + amylase-treated mice compared to the all the other groups (Figure 3C–G). To complement the in situ macrophages-related data, we performed a pathway analysis focused on 15 genes involved in the production and metabolism of nitric oxide, a component that is involved in inflammatory processes originating from macrophages. Among the analyzed genes, 12 exhibited an enrichment score significantly higher (one-way ANOVA, *p* < 0.05) in the probiotic + amylase group compared to the remaining three experimental cohorts (Appendix A). Moreover, we characterized the dendritic cell population to investigate which dendritic subset was more abundant in the mucosa of the probiotic + amylase-treated group. Evaluation of marker genes of different conventional dendritic cells (cDC)s indicated that cDCs type 1 were significantly more abundant, as highlighted by the *Cd8a* marker gene, but no difference was found related to the cDCs type2, as shown by the *Esam* marker gene not displaying significant difference among the four groups (one-way ANOVA test, *p* = ns). (Appendix A).

### 2.4. Probiotic Plus Amylase Blend Stimulates Apoptosis in Immune Cell Subsets

In order to highlight the signaling pathways that were inhibited or induced within the ROIs, we compared the three experimental groups (probiotics + amylase, only probiotics, and only amylase) with the control group (PBS) and charted gene expression values grouped by gene sets using volcano plots (Figure 4A). The resulting data show that signaling pathways in CD45^+^ cells were the most affected in the mice treated with the probiotic + amylase combination, showing 469 significantly differentially expressed pathways compared to the controls, while the probiotic-treated and the amylase-treated groups had 247 and 389 significantly differentially expressed pathways compared to the controls, respectively (Figure 4B,C). A complete list of significantly differentially expressed pathways among groups, including normalized enrichment score, adjusted *p* value, and target genes involved in each pathway, is presented in Appendix A. Interestingly, the lowest number of significantly differentially expressed pathways was found when comparing the “probiotics strains only” and the control groups, confirming the pivotal role of amylase in disrupting the biofilm present in the lumen of the intestine, allowing bacteria to infiltrate the ileal mucosa and promote immunomodulatory responses. Moreover, the comparison of the ROIs in the four experimental groups showed enrichment of five pathways related to apoptosis in the mice gavaged with the probiotics and amylase blend (Table 1). A heatmap of the apoptosis-related genes reveals a clear distinct expression only between the “probiotic + amylase group” compared to the control group. Specifically, among the 91 analyzed genes related to apoptosis, 70 of them showed a statistically significant differential expression in the “probiotic + amylase group” compared to the control within the selected ROIs of the two cohorts (Figure 4D). In comparison, only eight genes were differentially expressed between the probiotic-treated group and the control group (Figure 4E). Finally, 43 out of the 91 apoptosis-related genes analyzed were differentially expressed between the amylase-treated mice and the control group (Figure 4F), suggesting that amylase alone can breakdown biofilm, thus promoting the ability of commensal bacteria to infiltrate the intestinal mucosa and induce immune responses in the gut. A complete list of apoptosis-related genes differentially expressed in the ROIs of the three experimental cohorts vs. the control group is presented in Appendix A. 

Furthermore, we analyzed the expression of genes involved in the Toll-like receptors (TLR)s cascade pathways to highlight any differences among the four experimental cohorts, as TLRs are cell surface receptors considered to play a critical role in triggering innate immune responses by recognizing and binding to molecules present on microbes’ cell walls [24], hence affecting several aspects of IBD pathogenesis [25]. Seven out of nine pathways, except TLR4 and TLR9 cascade, were significantly increased in mice gavaged with probiotics and amylase compared to the control group. In contrast, mice gavaged with only probiotics displayed no statistical difference, except for TLR4 cascade, compared to the controls, confirming the pivotal role of amylase in enhancing the immunomodulatory effect of the probiotic strains. Moreover, mice gavaged with only amylase displayed significantly different pathways for seven out of the nine cascade pathways compared to controls, except for TLR7/8 and TLR9 cascade (Table 2). 

Lastly, considering that neutrophils are the most abundant immune cells present in the lamina propria during inflammation [26], and based on our GeoMx results indicating that neutrophils are the only immune subsets significantly decreased in the probiotic + amylase group compared to the other experimental groups, we evaluated the frequency of neutrophils undergoing the process of apoptosis. In order to do so, we performed fluorescence imaging on ileal tissues from our experimental groups with a deoxynucleotidyl transferase (TdT)- deoxyuridine triphosphate (dUTP)-biotin nick end labeling (TUNEL) assay to detect apoptosis and neutrophils staining. Our results clearly indicate an increased number of cells actively undergoing the apoptotic process (green) and fewer neutrophils (red) in the mucosa of mice gavaged with the probiotic and amylase mix compared to the control group (Figure 4G). These data suggest that probiotics are able to infiltrate the ileal mucosa, possibly thanks to the anti-biofilm activity of amylase, inducing a downstream immune response promoting apoptosis mostly in neutrophils.

Together, the GeoMx DSP and TUNEL assay data identify spatially distinct changes in immune subsets within distinct ROIs in the four experimental groups, particularly emphasizing differences related to apoptosis in immune cell subsets.

## 3. Discussion

In the present study, we performed transcriptome DSP analysis to investigate the interaction between a recently developed probiotic mix coupled with amylase and the immune cell subsets residing in the gut mucosa of the SAMP mouse model of CD-like ileitis. Initially, we showed that amylase and the probiotic blend are both necessary to ameliorate inflammation, since mice administered amylase alone or the probiotic strains mix alone did not show any significant amelioration compared to the PBS-administered control group. All the ROIs examined in this study were acquired in the mucosal layer and subsequently processed by excluding epithelial cells and including only CD45^+^/PanCK^−^ cells; therefore, we concluded that the genetic alterations caused by the amylase addition to the probiotic mix were specific to immune populations present in the subepithelial environment, mainly the lamina propria.

Our NanoString GeoMx data indicated that 453 cellular pathways were significantly differentially expressed in the lamina propria of the probiotic + amylase-treated mice compared to a group administered only probiotics, suggesting that amylase was directly responsible for disrupting the biofilm and eliminating microbial species residing inside it, thus removing a barrier at the interface between intestinal epithelium and the gut lumen. After penetrating the epithelial layer, the probiotic strains encounter the basement membrane, which represents a filter that can partially prevent the infection in deeper tissues, but its functional integrity can be disrupted by epithelial cells’ inflammation [27]. This way, the probiotics can reach the subepithelial tissues, where they finally interact with host defenses, mainly represented by phagocytic cells. As a result of the biofilm disruption, amylase facilitates the infiltration of the daily administered beneficial probiotic strains into the epithelium and into the subepithelial layers of the intestine, thus inducing an enhanced immunomodulatory response in mucosal immune cell subsets, resulting in decreased ileitis.

This evidence is particularly relevant considering that multiple microbiome studies have highlighted a cooperative interaction between different microbial populations resulting in biofilm development [28,29]. In particular, in a condition of dysbacteriosis, biofilm formation enables fungi and bacteria to escape immune detection [30] and increase the production of oxygen reactive species, leading to disruption of the gastrointestinal barrier and leaky gut [31]. In contrast, the number of significantly different pathways between probiotic + amylase-treated mice and the control group was 469, indicating almost no difference between the control and the probiotics-only groups, as also confirmed by histological analysis of the ileal tissues showing a similar degree of inflammation. 

Since interactions between the host immune system and the gut microbiome are partially mediated by TLRs and IBD patients exhibit significantly different expressions of TLRs compared to people not affected by IBD [32,33], we examined the TLR cascade signaling pathways. Analysis of the expression of genes involved in TLR cascade signaling revealed that amylase addition to the probiotics combination played a pivotal role in promoting the infiltration of probiotics in the lamina propria and consequently enhancing the activation of TLR cascade pathways in innate immune cells. Our data strongly suggest that the enhanced stimulation of TLR pathways was connected to the observed increased abundance of Tregs (Figure 3G), as TLRs not only exert their effect on innate immune cells but can also boost Treg immunosuppressive activity [34], preventing other T lymphocytes from functioning efficiently, hence promoting microbial tolerance and decreasing inflammatory activity [35]. The decreased abundance of Tregs in IBD patients causes a disequilibrium between effector T lymphocytes and Tregs which in turn leads to suppression of immune tolerance and an out-of-control progression of the inflammatory process [36]. These data confirm that diverse gut-residing populations can differently affect the expression of TLRs, impairing TLR’s ability to induce an anti-inflammatory response in IBD. TLRs have also been reported to recruit and stimulate dendritic cells to present antimicrobial antigens to T lymphocytes [37], hence representing a link between adaptive and innate immune systems [38]. Interestingly, our data collected from the probiotic + amylase group confirm a connection between increased expression of genes involved in TLR cascade signaling pathways and increased abundance of dendritic cells as well as Treg populations.

Conversely, a GeoMx DSP analysis focused on the expression of genes involved in the activation and signaling of NOD-like receptors, proteins that play a main role in detecting pathogens and initiating immune responses [39], did not show any statistical difference (normalized enrichment score < ±2; *p* > 0.05) among the four experimental cohorts (Appendix A).

Furthermore, GeoMx analysis showed that a significant alteration in the expression of 77% of the analyzed apoptosis-related genes was associated with decreased abundance of neutrophils and increased abundance of dendritic cells, ILCs, Tregs, and macrophages in the lamina propria of probiotic + amylase-treated mice compared to the control group. In comparison, the expression of only 8.8% of the apoptosis-related genes was significantly altered in the mice gavaged with only probiotics compared to the control group, with no observed difference in the abundance of neutrophils, dendritic cells, Tregs, ILCs, or macrophages. Conversely, expression of 47% of the analyzed apoptosis-related genes was significantly altered in the mice gavaged with only amylase compared to the control group, although no difference was observed in the abundance of neutrophils, dendritic cells, Tregs, ILCs, or macrophages. This amount of differentially expressed genes can be explained by the fact that amylase alone is capable of disrupting the biofilm present on the surface of the intestinal lumen, hence altering the abundance of microorganisms already present in the gut and their consequent interaction with the immune subsets present in the mucosa. Our hypothesis is corroborated by 16S rRNA analysis published by our group in a previous paper, indicating changes in relative bacterial abundance after probiotic and amylase treatment, specifically affecting species belonging to the genus *Lachnoclostridium* (increased) and species belonging to the family *Lactobacillaceae* (decreased) [22]. 

Interestingly, comparing the differentially expressed genes between the probiotic + amylase group and amylase-only group, we noticed a consistent increased expression in the probiotic + amylase mice of gene family members encoding for linker histone H1 proteins such as *H1f0*, *H1f1*, *H1f3*, and *H1f5*, which normally are responsible for chromatin condensation (Appendix A). Essential steps during early apoptotic process involve DNA cleavage and H1 histone-dependent induction of chromatin condensation [40]. Multiple studies have reported an increased expression of genes encoding for linker histone H1 proteins [41] during early apoptosis, only observed in correlation with the event of DNA fragmentation, thus possibly indicating a prerequisite for accessibility to DNA and endonuclease activity. Therefore, we speculate that the probiotic strains utilized in our experiment were responsible for inducing the expression of genes encoding for linker histone H1 proteins in mucosal immune cell subsets and, as a consequence, triggering a higher degree of programmed cell death, promoting resolution of inflammation downstream. The downregulated expression of genes encoding for linker histone H1 proteins observed in the amylase-only-treated group may account for the observed decreased apoptosis and the consequent similar level of inflammation compared to the control group. 

The probiotics strains used in our experiment have been documented in the literature as exhibiting pro-apoptotic activity towards neutrophils and other cell types. Specifically, Sustrova et al. demonstrated that *L. rhamnosus* exerts pro-apoptotic effects on human neutrophils in vitro [42], while *L. Acidophilus* has been reported to induce apoptosis in antigen-stimulated T lymphocytes [43]. Moreover, *S. boulardii* [44] and B. breve [45] have also been confirmed to have pro-apoptotic properties in vitro, as both probiotics have been tested on human cancer cells. Since dysregulated infiltration of neutrophils into the gut mucosal layer can enhance severe tissue damage, neutrophil apoptosis needs to be tightly regulated to maintain mucosal homeostasis [46]. A direct interaction between probiotic strains and immunocompetent cells in the gut-associated lymphoid tissue can promote and modulate apoptosis in immune cells [47,48]. This evidence clearly indicates that in our experiment, biofilm disruption caused by amylase allowed the probiotics strains to cause a higher degree of apoptosis in immune cell populations (particularly in neutrophils), facilitating resolution of inflammation. 

Moreover, DSP analysis showed that both *Caspase* (*Casp)3* and *Fas* genes were significantly upregulated in the probiotic + amylase-treated mice compared to the control group (Appendix A). Our data concurred with a previous study by Luo et al. [49] showing that decreased *Casp3* expression was correlated with delayed Fas-mediated apoptosis in neutrophils present in the umbilical cord, contributing to chronic inflammation. Besides *Casp3*, we also found increased *Casp7* expression in leukocytes of probiotic + amylase-treated mice compared to the remaining three groups. Our results (Appendix A) are in line with a study by Akhter et al. [50], showing that *Casp7^−/−^* mice harbor macrophages unable to inhibit the replication of intracellular bacterial pathogens such as *Legionella pneumophila* due to delayed induction of macrophage apoptosis, leaving them more susceptible to infection. The fact that both *Casp3* and *Casp7* were both consistently increased in the ROIs of the probiotic + amylase-treated mice compared to the other three groups is not surprising, since both genes code for proteases which have multiple endogenous substrates in common and are both activated during apoptosis by caspases-8 and -9 [51]. 

In addition, the data related to increased expression of *Tnfsf10*, *Tnfrsf10b*, *Ripk1*, *Fadd*, and *Casp8* in CD45^+^ cells suggest that programmed cell death was also triggered in immune cells through the RIPK1-dependent apoptosis (RDA) pathway. In fact, this pathway starts with activation of Tumor Necrosis Factor Receptor 1 (TNFR1), which in turn activates the formation of the TNFR1 complex, leading to the recruitment of multiple E3 ubiquitin ligases, which contribute to the ubiquitination of RIPK1. The subsequent deubiquitination of RIPK1 promotes the production of the pro-apoptotic complex-IIb formed by FADD, caspase-8, and RIPK1, which mediates the RDA [52]. These findings highlight the importance of the mucosal layer as a functional hotspot in IBD where distinct interactions between gut-residing microorganisms and innate immune cells may contribute to aggressive CD manifestation characterized by high relapse rate, need for recurrent surgeries, and penetrating disease [53]. Interestingly, a co-evolution theory between immune cells and their microenvironment has been already suggested [54], and our data support the idea that this phenomenon may occur preferentially in the lamina propria.

Finally, the TUNEL assay data coupled with IHC/IF staining confirmed the presence of an increased number of immune cells in the active phase of apoptosis and a decreased number of neutrophils in the mucosal layer of probiotic + amylase-treated mice compared to the control group, confirming our theory that amylase’s anti-biofilm activity allows probiotic strains to infiltrate deeper in the intestinal mucosa, triggering apoptosis in immune cells in the lamina propria, particularly affecting neutrophils.

## 4. Materials and Methods

### 4.1. Experimental Animals 

The SAMP mouse colony (species: *Mus musculus*) was maintained at Case Western Reserve University in the Animal Resource Center. The ileal phenotype of the CD-like ileitis SAMP mouse model occurs spontaneously and exhibits a time course that enables examination of the pre-disease, initial, and chronic stages of ileitis. Moreover, SAMP mice display notable similarities to CD in terms of histologic features, response to conventional CD therapies, and systemic symptoms [21]. Seven-week-old mice, sex- and age-matched between the experimental cohorts, were used to conduct the experiments. Mice were kept in micro-isolator cages (Allentown Inc., Allentown, NJ, USA) with 1/8-inch corn bedding and cotton nestlets used for environmental enhancement (Inotiv, West Lafayette, IN, USA). Mice had access to laboratory diet P3000 (Harlan Teklad, Indianapolis, IN, USA) and water during the whole experiments. All studies were performed in a blinded fashion, and mice were subjected to randomization and identified using a numerical code known only to the animal caretaker. The identification code was revealed only at the end of the experiments.

### 4.2. Test Materials and Administration Protocol

The probiotic strains and amylase enzyme were supplied by BIOHM Health, LLC (BIOHM Health, Cleveland, OH, USA). Mice were divided into four experimental cohorts and administered four different treatments through oral inoculation every day for 56 days: (1) probiotic + amylase blend (concentration: 0.25 mg/100 uL PBS); (2) only probiotics; (3) only amylase; and (4) PBS (vehicle-treated controls). The total concentration of the probiotic strains was 1 × 10^11^ colony forming units (CFU)s/g of lyophilized powder. Each probiotic strain was added in equal amounts: 0.25 × 10^11^ CFUs/g. The amylase was used at a concentration of 500 α-amylase dextrinizing units (SKB)/g.

### 4.3. Histology

Experimental mice were euthanized, and terminal ilea were collected, opened longitudinally, rinsed in PBS, and fixed in formalin. Tissues were then processed, embedded in paraffin, sectioned (5 μm thick), stained with hematoxylin and eosin (H&E), and histologically evaluated by a trained gastrointestinal pathologist in a blinded manner, using a validated semiquantitative scoring system as previously described [55]. Briefly, histology scores ranging from 0 (normal) to 3 (maximum level of changes) were used to analyze five histologic parameters: (1) villus distortion, (2) chronic inflammation (lymphocytes in the mucosal layer), (3) active inflammation (infiltration with neutrophils), (4) mononuclear inflammation (monocytes and macrophages), and (5) transmural inflammation.

### 4.4. GeoMx NanoString Data Processing and Analysis 

Paraffin-embedded slides were incubated at 65 °C for 2 h and then rehydrated in EtOH and ddH_2_O and treated with proteinase K (1.0 μg/mL at 37 °C for 15 min). Epitope retrieval (ER2 at 100 °C for 20 min) was also performed. Next, the sections underwent a process of hybridization with the CTA probes at 37 °C for 24 h. In order to identify tissue morphology landmarks, samples were washed for five minutes with buffer and formamide solution (1:1), and then, using the TME Morphology Kit (Manufacturer: NanoString Technologies, Seattle, WA, USA), they were stained for CD45 (NBP234528AF647, Novus Biologicals) and PanCK (NBP2-33200AF488, Novus Biologicals) to target immune and epithelial cells, respectively. Syto83 (S11364, Invitrogen) was used for nuclear detection. Next, the slides were loaded into the DSP machine and scanned for subsequent immunofluorescent imaging. Regions of Interests (ROI)s on the mucosal layer of the ilea were selected by a pathologist based on CD45 staining. The FASTQ files obtained through sequencing were processed using a GeoMx NGS Pipeline to generate gene count data for each ROI. Measurement of the number of CD45^+^ cells in the ROIs was performed using the DSP platform [56], and the acquired data were analyzed using the R software (version “4.4”) [57]. Graphs were created using the ComplexHeatmap (Version 2.22.0) package. Filtering and quality control were performed following the manufacturer’s protocol (NGS Data Analysis, MAN-10119-01). Spike-in probes for RNA targets were used to quantify the expression of 22,000 murine genes in a high-throughput manner from ileum. Next, the geometric mean for each target and the limit of quantification (LOQ: value obtained calculating two standard deviations above the negative control probe counts’ geometric mean) were quantified for each ROI. The next data filtration step consisted of excluding from subsequent analysis the target values that showed no measurement exceeding the LOQ value. The next step included a third quartile (Q3) normalization method performed following the GeoMx NanoString directions. Principal component analysis, heatmap dendrograms, and volcano plots representing pathways analysis were obtained using the custom scripts SpatialDecon_plugin.R and DimReduction.R. A cell profile matrix was derived from publicly available single cell RNA-sequencing “mouse cell atlas (MCA)” datasets [58] in order to facilitate immune cell deconvolution and differentiate between the immune cell populations in the mucosal layer of the intestine. Specifically, the dataset used was Mouse/Adult/SmallIntestine_MCA. We then calculated the abundance of every immune cellular population from the dataset employing the cellular-type categorization from the aforementioned paper [58].

### 4.5. TUNEL Assay

The degree of apoptosis in ileum tissues was evaluated utilizing the TUNEL technique. Tissue sections were double stained with anti-neutrophil antibody (Novus Biologicals LLC, Centennial, CO, USA) and the Click-iT^TM^ Plus TUNEL Assay kit (Thermo Fisher Scientific, Waltham, MA, USA) was used following the manufacturer’s instructions.

### 4.6. Statistical Analysis

Experiments were executed and repeated in duplicate, and the acquired data were used in multivariate analyses. Unpaired Student’s t test was used to compare data obtained from the experimental mouse cohorts and determine possible differences. Data are represented as mean ± standard error of measurements (SEM)s, with 95% confidence intervals being reported. An alpha level value of 0.05 was regarded as significant. GraphPad software (San Diego, CA) (Version 10.4.0) was used to perform the analyses..

## 5. Conclusions

In conclusion, our data demonstrate a beneficial role of the recently developed probiotic and amylase blend in ameliorating ileitis in the SAMP mice. The main beneficial effect was derived from an increased induction of apoptosis in neutrophils present in the lamina propria, which in turn facilitate the resolution of inflammation. Chronic non-resolving intestinal mucosal inflammation, involving innate immune responses, plays a crucial role in IBD pathophysiology [59], suggesting that the recently developed probiotic + amylase blend is ready for testing in CD clinical trials.

## Figures and Tables

**Figure 1 ijms-25-12066-f001:**
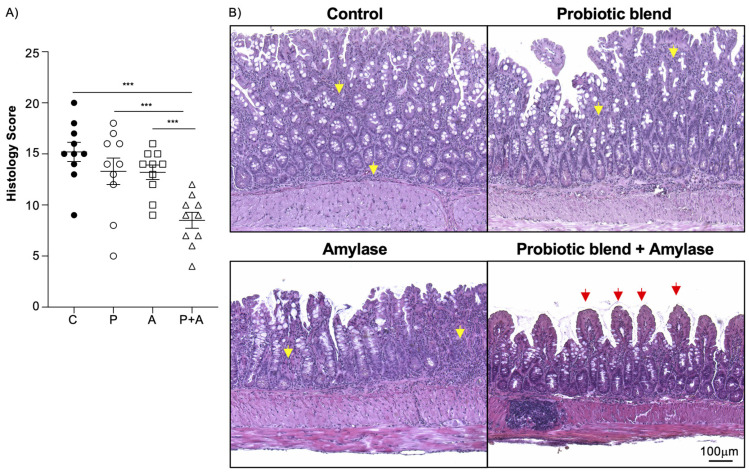
Amylase and probiotics are both necessary to ameliorate inflammation in SAMP mice. (**A**) Histology evaluation shows significant attenuation of ileitis in mice treated with amylase plus probiotic mix (P + A) in comparison with the mice treated with only amylase (A), only probiotic mix (P), or PBS (control) (C) (one-way ANOVA, 8.50 ± 2.46 vs. 13.20 ± 2.25 vs. 13.30 ± 4.08 vs. 15.20 ± 2.97; *p* < 0.001). (**B**) Representative pictures of H&E-stained sections indicate that mice treated with both amylase and probiotic mix have better-preserved architecture of the villi (red arrows) and less presence of inflammatory cells in the mucosal and submucosal layer (yellow arrows) compared to the other three groups. Data are represented as mean ± SEM and are representative of two independent experiments; *** *p* < 0.001. N = 10/group. 10X + 1.25 original mag.

**Figure 2 ijms-25-12066-f002:**
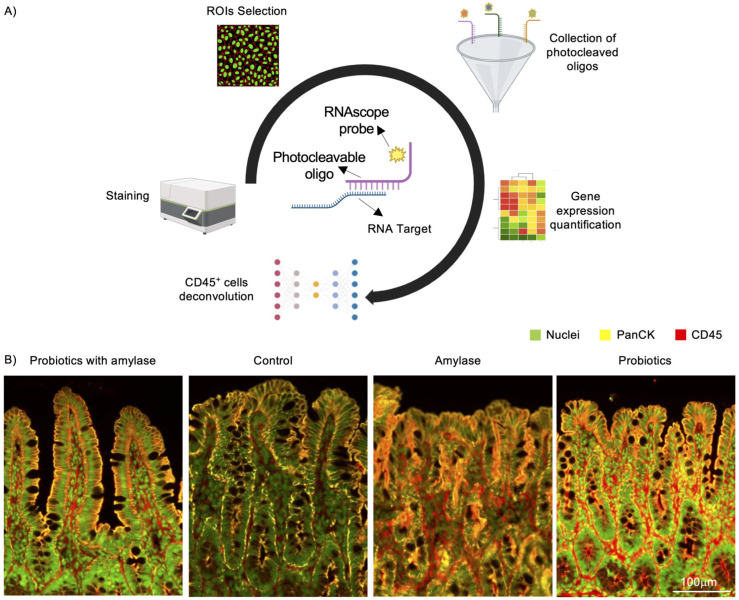
Digital spatial profiling of genetic expression performed on mucosal ileal tissue in SAMP mice. (**A**) Schematic workflow showing immunohistochemistry (IHC)/immunofluorescence (IF) staining of slides embedded in paraffin with markers for PanCK, CD45, and nuclei in the ROIs. (**B**) Representative images of mucosal layers of stained tissues with segments superimposed and with probe counts indicating PanCK^+^ cells (Cy3568 nm; yellow), CD45^+^ cells (Texas red, 615 nm; red), and nuclear staining (FITC, 525 nm; green). N = 6 mice/group; N = 6 ROIs/mouse.

**Figure 3 ijms-25-12066-f003:**
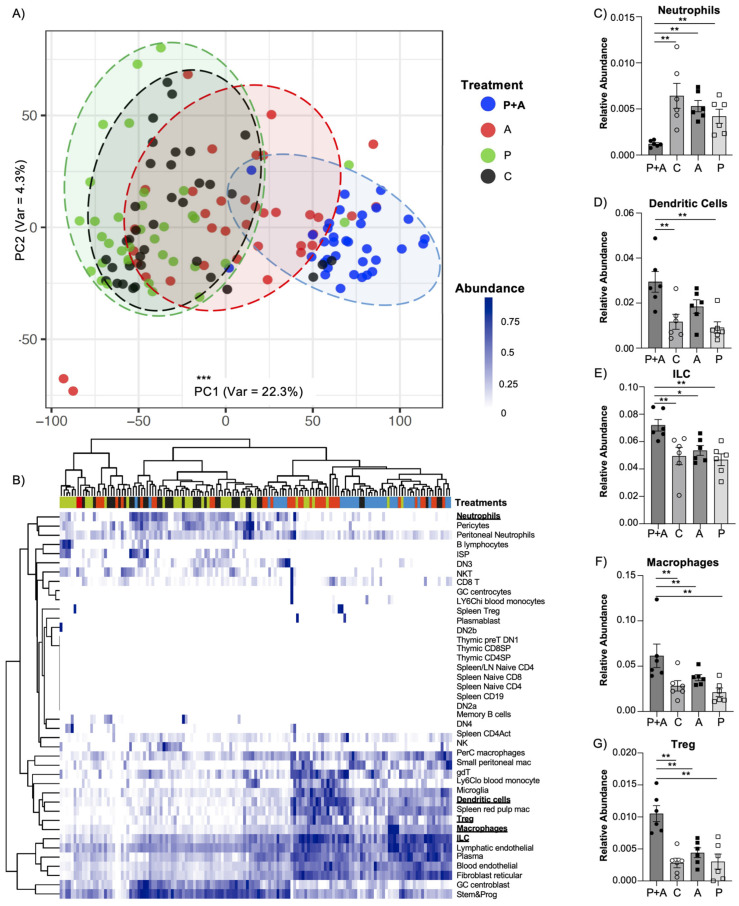
Probiotics mix coupled with amylase exerts an enhanced immunomodulatory effect in the mucosal layer. (**A**) Principal component analysis showing the variance among ROIs based on cell type abundance beta estimates (PC1 variance: 22.3%; PC2 variance: 4.3%), highlighting significant difference in the mice administered the probiotics + amylase mix (P + A) compared to only amylase (A), only probiotics (P), and control (C) groups (PC1: Kruskal–Wallis test, 69.94 ± 3.94 vs. 5.39 ± 7.35 vs. −42.63 ± 5.89 vs. −30.00 ± 5.56; *p* < 0.001; N = 36 ROIs/group). (**B**) Dendrogram showing estimated relative abundance of immune cell subsets in microenvironment segments, indicating a significant decrease in (**C**) neutrophils (one-way ANOVA, 1.22 × 10^−3^ ± 0.14 × 10^−3^ vs. 6.43 × 10^−3^ ± 1.35 × 10^−3^ vs. 5.30 × 10^−3^ ± 0.61 × 10^−3^ vs. 4.21 × 10^−3^ ± 0.76 × 10^−3^; *p* < 0.02) and increase in (**D**) dendritic cells (one-way ANOVA, 2.29 × 10^−2^ ± 0.46 × 10^−2^ vs. 1.12 × 10^−2^ ± 0.33 × 10^−2^ vs. 0.91 × 10^−2^ ± 0.25 × 10^−2^; *p* < 0.02), (**E**) innate lymphoid cells (ILC)s (one-way ANOVA, 7.19 × 10^−2^ ± 0.42 × 10^−2^ vs. 4.94 × 10^−3^ ± 0.63 × 10^−2^ vs. 5.35 × 10^−2^ ± 0.36 × 10^−2^ vs. 4.68 × 10^−2^ ± 0.43 × 10^−2^; *p* < 0.02), (**F**) macrophages (one-way ANOVA, 6.15 × 10^−2^ ± 1.29 × 10^−2^ vs. 2.28 × 10^−2^ ± 0.58 × 10^−2^ vs. 3.73 × 10^−2^ ± 0.34 × 10^−2^ vs. 2.21 × 10^−2^ ± 0.46 × 10^−2^; *p* < 0.02), and (**G**) regulatory T cells (Treg)s (one-way ANOVA, 1.05 × 10^−2^ ± 0.12 × 10^−2^ vs. 0.80 × 10^−2^ ± 0.07 × 10^−2^ vs. 0.44 × 10^−2^ ± 0.08 × 10^−2^ vs. 0.30 × 10^−2^ ± 0.12 × 10^−2^; *p* < 0.02) in the group treated with probiotic + amylase compared with control, only amylase, and only probiotics groups. Data are represented as mean ± SEM and are representative of two independent experiments. * *p* < 0.05; ** *p* < 0.02; *** *p* < 0.001. N = 6 mice/group; N = 6 ROIs/mouse.

**Figure 4 ijms-25-12066-f004:**
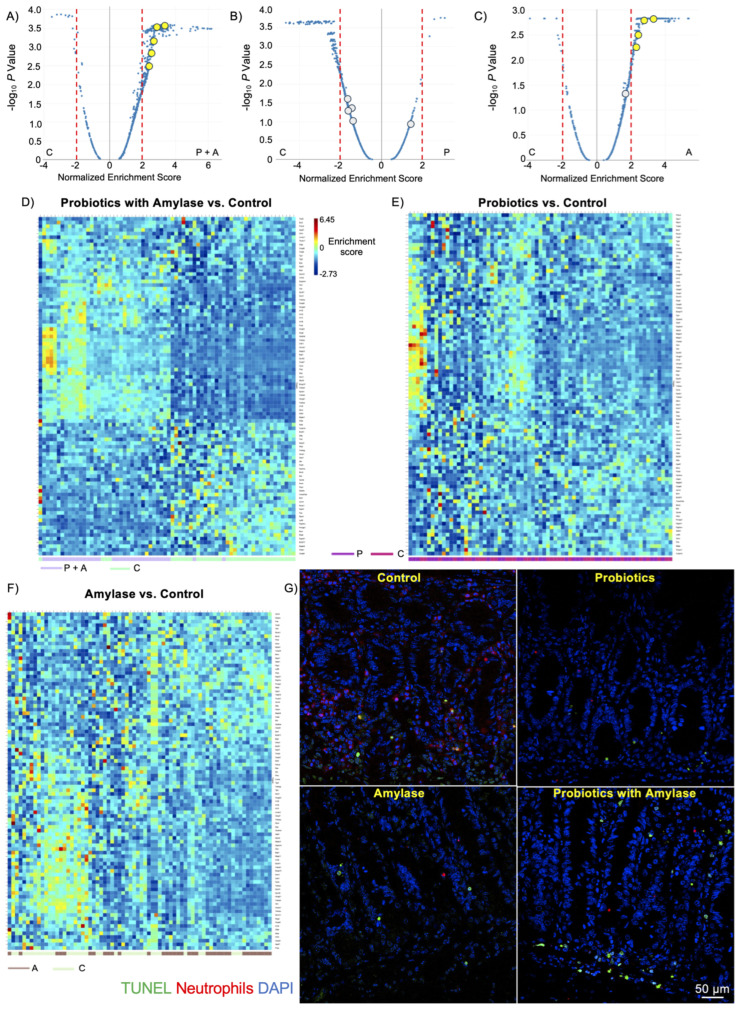
Probiotic + amylase blend stimulates apoptosis in immune cell subsets. (**A**–**C**) Volcano plots obtained from the pathway analysis of CD45^+^ cells in ROIs of the four experimental groups. Statistically significant different apoptosis-related pathways are labeled in yellow (*p* < 0.05; normalized enrichment score > ±2) and in gray when not significantly different. Amylase and the probiotic mix are both necessary to significantly induce apoptosis in CD45^+^ cells. Comparison between probiotics without amylase and the control group shows no significantly different apoptosis-related pathways between the two groups. (**D**) Heatmap of differentially expressed genes related to apoptosis pathways between the control group and probiotic + amylase, (**E**) only probiotics, and (**F**) only amylase groups. (**G**) Representative IF photomicrographs of TUNEL assay show the increased number of apoptotic TUNEL-positive cells (green, Alexa Fluor™ 488) and fewer neutrophils (red, Alexa Fluor™ 594) deep in the lamina propria of mice treated with the probiotic + amylase blend compared to the control group (DAPI, blue). 40X original mag. N = 6 mice/group.

**Table 1 ijms-25-12066-t001:** Comparison of apoptosis-related pathways among the experimental groups. All five pathways are significantly increased (normalized enrichment score > ±2) in mice gavaged with probiotics and amylase (P + A) compared with mice gavaged with only probiotics (P), only amylase (A), or PBS (C).

Normalized Enrichment Score	P + A vs. C	P vs. C	A vs. C	P + A vs. P	P vs. A	P + A vs. A
Apoptosis	3.08	1.87	3.51	2.93	3.61	2.69
Apoptotic Cleavage of Cellular Proteins	2.61	1.45	2.28	2.66	2.65	2.56
Apoptotic Execution Phase	3.20	1.46	3.38	3.16	3.49	2.96
Apoptotic Factor-Mediated Response	2.22	−0.71	1.69	2.15	1.67	2.22
Intrinsic Pathway of Apoptosis	2.56	1.17	2.57	2.46	2.71	2.18

**Table 2 ijms-25-12066-t002:** Comparison of TLR cascade pathways among the experimental groups. Seven out of nine pathways, except TLR4 and TLR9 cascade, are significantly increased (normalized enrichment score > ±2) in mice gavaged with probiotics and amylase (P + A) compared with controls (C). Mice gavaged with only probiotics (P) display no statistical difference, except for the TLR4 cascade, compared to the controls. Mice gavaged with only amylase (A) display significantly different pathways for seven out of the nine cascade pathways, except for TLR7/8 and TLR9 cascade, compared to controls.

Normalized Enrichment Score	P + A vs. C	P vs. C	A vs. C	P + A vs. P	P vs. A	P + A vs. A
TLR1:TLR2 Cascade	2.08	−1.86	2.25	2.04	−2.45	2.14
TLR2 Cascade	2.08	−1.86	2.25	2.04	−2.45	2.14
TLR3 Cascade	2.29	−1.86	2.33	2.22	−2.49	2.41
TLR4 Cascade	1.96	−2.05	2.33	1.90	−2.68	2.07
TLR5 Cascade	2.13	−1.92	2.20	2.08	−2.41	2.21
TLR6:TLR2 Cascade	2.08	−1.86	2.25	2.04	−2.46	2.14
TLR7/8 Cascade	2.02	−1.98	1.96	2.01	−2.25	2.10
TLR9 Cascade	1.90	−1.91	1.85	1.89	−2.19	1.99
TLR10 Cascade	2.13	−1.92	2.20	2.08	−2.41	2.21

## Data Availability

The data that support the findings of this study are available from the corresponding author, LDM, upon reasonable request. Data will be stored for a long-term period (minimum 5 years) in the Box storage service (hosted in the cloud) that enables Case Western Reserve University to store, access, and share files securely. Box is the only approved platform for storing restricted data in the cloud at Case Western Reserve University.

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
