# Peer review of "A New Probiotic Formulation Promotes Resolution of Inflammation in a Crohn’s Disease Mouse Model by Inducing Apoptosis in Mucosal Innate Immune Cells"

_ijms, 2024, doi:10.3390/ijms252212066_

Round 1
Reviewer 1 Report
Comments and Suggestions for Authors
In the manuscript entitled “ A new probiotic blend facilitates resolution of inflammation in 2 a Crohn’s Disease mouse model by promoting apoptosis in mucosal innate immune cells”, De Salvo and colleagues aimed to investigate if a probiotic and amylase blend decrease ileitis in the SAMP mouse models.
The authors report that the administrated blend promote significant alterations in dendritic cells, innate lymphoid cells, macrophages and neutrophils. Additionally, suggest that the administrated probiotic can infiltrate into the intestinal mucosa, triggering apoptosis of lamina propria neutrophils, facilitating resolution of inflammation in the ileum.
While the study presented is of interest there are some points that needs to be carefully addressed.
The experimental design is poorly explained and no indication of how many CFU of probiotic were administrated.
According to the manuscript the statistical analysis in Fig. 1A and Fig. 3C-F has been done with the Unpaired Student’s t test. However to compare three groups or more, an ANOVA test should be performed.
Fig.3B cannot be read and is poorly explained.
Fig.3C-F Authors should indicate how they differentiate between the different cell populations
Fig. 4D-F is impossible to read properly, please change it.
Authors focused on neutrophils because they were reduced after the probiotic + amylase treatment compare to the other groups. Nevertheless, dendritic cells, ILCs and macrophages were significantly increased in the probiotic + amylase treatment compare to the other groups. However, no further explanation/analysis has been performed. It could have been of importance to analyse the phenotype of both, macrophages and dendritic cells increased.
Author Response
The experimental design is poorly explained and no indication of how many CFU of probiotic were administrated.
The section 4.2 (now section 5.2) describing the experimental design has been updated (lines 429-437). The total concentration of the probiotic strains was 1x1011 colony forming units (CFU)s/g of lyophilized powder. Each probiotic strain was added in equal amount: 0.25x1011 CFUs/g for each strain. The amylase was used at a concentration of 500 α-amylase dextrinizing units (SKB)/g (lines 434-437).
According to the manuscript the statistical analysis in Fig. 1A and Fig. 3C-F has been done with the Unpaired Student’s t test. However, to compare three groups or more, an ANOVA test should be performed.
The Unpaired Student’s t test has been replaced with the One-way ANOVA test in Figure 1 (line 112) and Figures 3C-F (lines 168, 169, 171 and 172).
Fig.3B cannot be read and is poorly explained.
The size of Figure 3B has been increased (line 160), and the explanation has been clarified and improved (lines 140-144).
Fig.3C-F Authors should indicate how they differentiate between the different cell populations
A cell profile matrix was derived from publicly available single cell RNA-sequencing “mouse cell atlas (MCA)” datasets (PMID: 29474909) in order to facilitate immune cell deconvolution and differentiate between the immune cell populations in the mucosal layer of the intestine. Specifically, the dataset used was Mouse/Adult/SmallIntestine_MCA. We then calculated the abundance of every immune cellular population from the dataset employing the cellular-type categorization from the aforementioned paper. An explanation has been added in the Section 5.4 (lines 474-480).
Fig. 4D-F is impossible to read properly, please change it.
Since listing all the genes in the heatmaps at a minimum font’s size of 6 is unfeasible to maintain all the data in the same figure, the complete list of genes differentially expressed in the ROIs of the three experimental cohorts vs. the control group is now present in Figure S3. Please see Figures 4D-F’s where the size have been increased (line 236).
Authors focused on neutrophils because they were reduced after the probiotic + amylase treatment compared to the other groups. Nevertheless, dendritic cells, ILCs and macrophages were significantly increased in the probiotic + amylase treatment compared to the other groups. However, no further explanation/analysis has been performed. It could have been of importance to analyze the phenotype of both, macrophages and dendritic cells increased.
To complement the macrophages-related data, we evaluated the expression of 15 genes involved in the metabolism of nitric oxide, a component that is involved in inflammatory processes originating from macrophages. Among the analyzed genes, twelve exhibited an enrichment score significantly higher (one-way ANOVA test, P<0.05) in the probiotic+amylase group compared to the remaining three experimental cohorts (Figure S1) (lines 148-154). Moreover, we characterized the dendritic cell population to investigate which dendritic subset was more abundant in the mucosa of the probiotic+amylase treated group. Evaluation of marker genes of different conventional Dendritic cells (cDC)s highlighted that cDCs type 1 were significantly more abundant, as highlighted by the Cd8a marker gene (one-way ANOVA test, P<0.05), but no difference was found related to the cDCs type2, as shown by the Esam marker gene not displaying significant difference among the four groups (one-way ANOVA test, P>0.05). (Figure S1) (lines 154-160).
Reviewer 2 Report
Comments and Suggestions for Authors
This article relates to a field of a previous communication research which is currently under review (ijms-3272888), written by partly overlapping authors.
In the current article, the authors aim to investigate the immunomodulatory role of a newly developed probiotic+amylase complex in a SAMP mouse model. In mice given probiotics and amylase, DSP analysis revealed a high enrichment of five intracellular pathways linked to apoptosis, along with a decrease in neutrophil number and a rise in dendritic cells, innate lymphoid cells, and macrophages. The TUNEL assay confirmed the increased prevalence of apoptosis. These findings show that the probiotic and amylase blend has a positive effect on inflammation resolution by increasing the apoptosis of cell subsets linked to innate immunity.
The study is of significant clinical importance. It is well-designed and well-presented.
The results are clear, and the figures and tables all help the understanding of the results.
The methods are adequate, and their description is clear.
The discussion lacks an in-depth molecular explanation of the results.
What could be the molecular explanation for the synergistic effect of the added amylase? On what cell types does it display its main effect? On an epithelial layer? Or on the subepthelial layer? In the latter, then how? To reach the subeptithelial immune cells, colleganese is usually used.
If probiotics were used by the authors, could there be a connection between probiotics and immune cell apoptosis? Is the association solely with RAD apoptosis and H1 proteins? Are the Toll-like and NOD-like receptors included?
I recommend broadening the discussion by incorporating these aspects.
Author Response
This article relates to a field of a previous communication research which is currently under review (ijms-3272888), written by partly overlapping authors.
We acknowledge that the topic of the two papers seem to partially overlap. However, the referred article (ijms-3272888) is focused on a different model (Dextran Sodium Sulfate (DSS)-induced inflammation), and the colon is the organ analyzed. The model described is chemically-induced, while our model describes a spontaneous phenotype on the ileal inflammation. Moreover, in our paper we describe the use of cutting-edge GeoMx technique to analyze how a novel probiotic-based treatment can affect ileitis and the gene expression of immune cells specifically present in the mucosal layer of the ileum. In the aforementioned paper (ijms-3272888), the authors analyzed the effect of a particular amino acid (leucine) on the chemically-induced colitis model. The tools used to characterize the colitis levels were: myeloperoxidase assay and colonoscopy. Both methods are not part of the current manuscript, since colonoscopy cannot be performed in the ileum (focus of our manuscript), but only in the colon (focus of their manuscript). The partial overlapping of the authors is due to the fact that I (Luca Di Martino) was asked to collaborate with other members of Case Western Reserve University, since I am the only scientist in Ohio with expertise in performing colonoscopy on mouse models and evaluating degree of inflammation. Moreover, the referred article is a “communication” article type (less than 2,000 words), while our manuscript is a complete research article with more insight into the mechanism underlying the effect of the Probiotic + amylase blend (more than 5,000 words).
The discussion lacks an in-depth molecular explanation of the results.
What could be the molecular explanation for the synergistic effect of the added amylase? On what cell types does it display its main effect? On an epithelial layer? Or on the subepithelial layer? In the latter, then how? To reach the subepithelial immune cells, collagenase is usually used.
We thank the Reviewer for the suggestions. The synergistic effect of adding amylase has been now added in the discussion (lines 274-287). Amylase’s activity on the immune cells is indirect, since it removes a physical barrier (biofilm) containing pathogens and commensal bacteria, facilitating the infiltration of the probiotic strains into the epithelium and into the subepithelial layers of the intestine, mainly affecting the abundance of: neutrophils (lines 221-233, 363-370), regulatory T cells (lines 303-310) and dendritic cells (lines 154-160, 313-317).
If probiotics were used by the authors, could there be a connection between probiotics and immune cell apoptosis?
The discussion section was expanded to include aspects related to the connection between probiotics and immune cell apoptosis (lines 357-370).
Is the association solely with RAD apoptosis and H1 proteins? Are the Toll-like and NOD-like receptors included?
I recommend broadening the discussion by incorporating these aspects.
The discussion has been broadened incorporating aspects related to Toll-like receptors (lines 297-317). Additionally, a table related to TLR signaling pathways has been added (lines 255-261) and described in the “Results” section (lines 209-220).
GeoMx DSP analysis focused on the expression of genes involved in NOD-like receptors pathways did not show any statistical difference (normalized enrichment score<+2) among the four experimental cohorts (lines 318-322).
Reviewer 3 Report
Comments and Suggestions for Authors
REVIEW
Dear authors,
The work presents an alternative treatment to regulate the inflammatory process caused by Crohn's disease in a murine model (SAMP) using a mixture of probiotic microorganisms and the enzyme amylase, obtaining favorable results where the apoptosis of neutrophils involved in inflammation was promoted, reestablishing the normal architecture of the intestinal epithelium in the treated mice at the end of the evaluation. However, I consider that corrections should be made to improve the quality of the work.
Please amend the requested comments and submit the revision file.
1. Section 4.1 Experimental Animals lacks information on the mouse strain used (SAMP), although previous works mention the use of this strain, I consider it necessary to describe the particular characteristics of the animal model used to develop Crohn's disease-like ileitis.
2. Is any treatment necessary for SAMP mice to develop ileitis? If so, please mention this and describe the treatment used.
3. They do not indicate the sex of the SAMP mice and, more importantly, the total number of mice used in the experiment, as well as those used in each group.
4. In section 4.2 Test Materials and Administration Protocol they indicate that they used a concentration of the probiotics+amylase mixture of 0.25 mg/100 uL PBS, however, they do not mention the CFU concentration of each probiotic microorganism or the amount of the amylase enzyme. It is necessary to know the quantities because there are groups where they only administered the probiotic mixture and the amylase enzyme individually.
5. Were the different treatments administered by oral inoculation or in the drinking trough?
6. How did they determine the administration time of the 3 treatments (56 consecutive inoculations)?
7. In section 2.1 Amylase and probiotics are both necessary to ameliorate inflammation in SAMP mice, they describe the methodology again (lines 100-103).
8. In Figure 1A, the abbreviations C, P, A and P+A are missing in the figure caption.
9. Section 5. Conclusions is missing.
10. The results show that the treatment induces apoptosis of the neutrophil population, however, is the effect produced directly by any of the probiotic strains administered or do they stimulate any of the indigenous microorganisms in the microbiota of the SAMP mice?
11. The effect of amylase administration is to degrade the intestinal mucosa to allow the stimulation of probiotics for in situ immune regulation, however, it must be considered that the formation of biofilm by the microbiota itself is a protection mechanism against pathogens and eliminating this barrier can cause the translocation of commensal enterobacteria to the lumen and consequently, exposure to antigens such as LPS, which is highly immunogenic. How would these results be extrapolated to a clinical phase in humans?
12. I consider that they should complement the in situ study with an evaluation of reactive nitrogen species such as nitric oxide, a component that is involved in inflammatory processes originating from macrophages, which increase when treatment with the mixture of probiotics and amylase is administered (Figure 3F).
13. As is already known, immunomodulation is not only in situ, but at a systemic level, so the determination of an inflammatory cytokine profile in serum would be necessary to consider that the treatment can immunomodulate the entire organism of the inflammatory process of ileitis.
14. Increasing the size of Figure 3B and Figure 4G makes them more difficult to observe.
Please amend the requested comments and submit the revision file.

Author Response
- Section 4.1 Experimental Animals lacks information on the mouse strain used (SAMP), although previous works mention the use of this strain, I consider it necessary to describe the particular characteristics of the animal model used to develop Crohn's disease-like ileitis.
We agree with the Reviewer. In this revised manuscript we updated Section 4.1 (now section 5.1) with more information about the characteristics of the SAMP mouse strain (lines 417-421).
- Is any treatment necessary for SAMP mice to develop ileitis? If so, please mention this and describe the treatment used.
One sentence has been added (lines 76-78) explaining that the SAMP mouse strain develops CD-like ileitis spontaneously, without chemical or immunological manipulation.
- They do not indicate the sex of the SAMP mice and, more importantly, the total number of mice used in the experiment, as well as those used in each group.
Mice were sex-matched in all the experiments (50% males; 50% females) (line 421). The number of mice in each experiment has been added in each Figure caption. Figure 1 (line 117), figure 2 (line 135), Figure 3 (lines 177-178), figure 4 (line 249).
- In section 4.2 Test Materials and Administration Protocol they indicate that they used a concentration of the probiotics+amylase mixture of 0.25 mg/100 uL PBS, however, they do not mention the CFU concentration of each probiotic microorganism or the amount of the amylase enzyme. It is necessary to know the quantities because there are groups where they only administered the probiotic mixture and the amylase enzyme individually.
The section 4.2 (now section 5.2) has been updated (lines 429-437). The total concentration of the probiotic strains was 1x1011 colony forming units (CFU)s/g of lyophilized powder. Each probiotic strain was added in equal amount: 0.25x1011 CFUs/g. The amylase was used at a concentration of 500 α-amylase dextrinizing units (SKB)/g (lines 434-437).
- Were the different treatments administered by oral inoculation or in the drinking trough?
Mice were administered four different treatments through oral inoculation (lines 431-432).
- How did they determine the administration time of the 3 treatments (56 consecutive inoculations)?
To determine the length of treatment time necessary to successfully treat the animals, we originally treat the mice for 4 weeks (28 consecutive days). At that time no significant differences between the groups was observed. Therefore, we extended the treatment to 8 weeks (56 days). Treatment for 56 days led to a significant increase in the efficacy of the probiotic+ amylase blend.
- In section 2.1 Amylase and probiotics are both necessary to ameliorate inflammation in SAMP mice, they describe the methodology again (lines 100-103).
The repeated sentence in section 2.1 has been now removed (line 103).
- In Figure 1A, the abbreviations C, P, A and P+A are missing in the figure caption.
The abbreviations have been added (lines 111-112).
- Section 5. Conclusions is missing.
A conclusion section has been added (lines 406-413).
- The results show that the treatment induces apoptosis of the neutrophil population, however, is the effect produced directly by any of the probiotic strains administered or do they stimulate any of the indigenous microorganisms in the microbiota of the SAMP mice?
Our results suggest that the anti-inflammatory effect exerted by the probiotics is obtained by an indirect effect, as the probiotic strains stimulate particular indigenous microorganisms in the SAMP microbiome, as previously described (PMID: 36715169). In particular,16s rRNA analysis of the fecal microbiome revealed a diverse microbial composition between probiotic+amylase-treated versus PBS-treated control mice. Specifically, there was no difference in the a- and b-diversity between the two groups prior to probiotic administration (i.e., at baseline). In contrast, following probiotic administration, a significant shift of the microbial population in probiotic-treated mice was observed. Multiple probiotic studies have demonstrated a positive correlation between increased richness of the gut microbiome and amelioration of symptoms in several disease states. These observations convincingly support the concept that the intestinal microbial alteration is possible through probiotic administration, and suggest that the main driving force for the decreased level of ileitis in probiotic-treated mice is through an indirect probiotic effect. In addition to the aforementioned variation in a- and b-diversity, we found significant changes in specific bacterial taxa, such as bacteria belonging to the genus Lachnoclostridium. Specifically, Lachnoclostridium bacteria were consistently increased in stool samples upon completion of the probiotic+amylase treatment compared to the PBS-treated control group. The Lachnoclostridium genus includes bacteria such as Clostridium XIVa, which is known to constitute a significant part of the human gut microbiome; it plays a role in homeostasis and can stimulate anti-inflammatory effects.
- The effect of amylase administration is to degrade the intestinal mucosa to allow the stimulation of probiotics for in situimmune regulation, however, it must be considered that the formation of biofilm by the microbiota itself is a protection mechanism against pathogens and eliminating this barrier can cause the translocation of commensal enterobacteria to the lumen and consequently, exposure to antigens such as LPS, which is highly immunogenic. How would these results be extrapolated to a clinical phase in humans?
We agree with the Reviewer that eliminating the biofilm leads to translocation of commensal bacteria to the epithelial surface, causing highly immunogenic response. As confirmation of this observation, our data collected from the group administered only amylase (without probiotic mix) showed no signs of amelioration related to ileitis compared to the control group. Our results indicate that certain probiotic strains found elevated in healthy first-degree relatives of Crohn’s disease (CD) patients have an anti-inflammatory effect on immune cells present in the subepithelial tissue. Specifically, apoptosis is highly induced in immune cells in the lamina propria, facilitating resolution of inflammation and consequently leading to amelioration of ileitis in comparison with the other 3 experimental cohorts. Since CD patients exhibit delayed apoptosis of intestinal neutrophils which can lead to persistence of the inflammatory response associated with inflammatory bowel disease (IBD) (PMID: 10807010), the data collected from this study will move our work into translational approach and provide the rationale for clinical testing of the designed probiotics-amylase combination as a supplement to existing main therapies used to keep CD patients in a phase of remission and consequently enhance the effects of the currently available IBD treatments. The initial goal of this clinical phase will focus on replicating observations taken from the SAMP mouse model (data collected from this manuscript and previous one (PMID: 36715169) into humans. Specifically, CD patients will be administered probiotic+amylase, and 16S rRNA will be performed on stool samples collected before and after the probiotic+amylase treatment to highlight any gut microbiome alterations. Moreover, the gut microbiome effects on intestinal immune cells will be investigated in biopsies collected from IBD patients by analyzing immune alterations in each distinct layer (mucosa, submucosa, muscularis and serosa) through the use of cutting edge GeoMx DSP technique (described in this manuscript). These data will provide information for a proof-of-concept clinical trial (second goal) focused on testing probiotics and amylase as adjuvant therapies combined with established IBD treatments (i.e., anti-TNF, immunosuppressants) to help maintain gut homeostasis in CD patients, thus decreasing the risk of flare-ups.
- I consider that they should complement the in situstudy with an evaluation of reactive nitrogen species such as nitric oxide, a component that is involved in inflammatory processes originating from macrophages, which increase when treatment with the mixture of probiotics and amylase is administered (Figure 3F).
The in-situ study has been complemented with the analysis of the enrichment score related to 15 genes involved in the metabolism of nitric oxide (lines 148-154). Among the analyzed genes, twelve exhibited an enrichment score significantly higher (one-way ANOVA <0.05) in the probiotic+amylase group compared to the remaining three experimental cohorts (Figure S1).
- As is already known, immunomodulation is not only in situ, but at a systemic level, so the determination of an inflammatory cytokine profile in serum would be necessary to consider that the treatment can immunomodulate the entire organism of the inflammatory process of ileitis.
We agree with the Reviewer that a determination of an inflammatory cytokine profile in serum would determine if the treatment can have immunomodulation at a systemic level. Unfortunately, we do not currently have serum samples from the experimental mice used in this manuscript. Obtaining the serum samples would entail not only to repeat the experiment (timeline: 56 days), but also wait for a group of sex-matched SAMP mice to reach the proper age (7-week-old) to start the experiment. We believe that the analysis of the systemic effect of the probiotic+amylase treatment goes beyond the scope of this paper, mainly focused on the in-situ effect of the probiotic combination
- Increasing the size of Figure 3B and Figure 4G makes them more difficult to observe.
Figure 3B (line 160) and Figure 4G’s size (line 236) have been increased
Round 2
Reviewer 1 Report
Comments and Suggestions for Authors
In the revised version of the manuscript, the authors have adequately addressed all the comments expressed by this reviewer.
Reviewer 2 Report
Comments and Suggestions for Authors
The authors correctly revised the manuscript which is now acceptable for publication.
Reviewer 3 Report
Comments and Suggestions for Authors
Dear authors,
The requested modifications were made, as well as the justification for the questions, so I consider that the work has the quality and information required for its publication.